# Effect of Silicon on Oat Salinity Tolerance: Analysis of the Epigenetic and Physiological Response of Plants

**Barbara Stadnik** [1,2,]*[ID]**, Renata Tobiasz-Salach** [2] **and Marzena Mazurek** [3][ID]

1    Doctoral School, University of Rzeszow, Rejtana 16C, 35-959 Rzeszow, Poland
2    Department of Crop Production, University of Rzeszow, Zelwerowicza 4, 35-601 Rzeszow, Poland
3    Department of Physiology and Plant Biotechnology, University of Rzeszow, Ćwiklińskiej 2, 35-601 Rzeszow, Poland
*    Correspondence: barbarast@dokt.ur.edu.pl

**Abstract:** Environmental conditions are the primary factor determining the growth and yield of plants. As a result of climate change, the negative impact of abiotic factors is intensifying. One of them is salt stress. Soil salinity is one of the major problems in agriculture in the world and affects many cultivar species. The aim of this study was to evaluate the effect of silicon foliar application on the physiological and epigenetic reaction of oats (*Avena sativa* L.) under salt stress. The pot experiment was carried out in controlled conditions. Oat plants were subject to sodium chloride (NaCl) at a concentration of 200 mM and applied to the soil. Three concentrations of Optysil (200 g·L$^{-1}$ SiO$_2$) were used for foliar fertilization. Measurements were made of the relative chlorophyll content in the leaves, the selected chlorophyll fluorescence parameters, and the gas exchange parameters. In this study, methylation-sensitive amplification polymorphisms (MSAP) analysis was used to investigate the effect of Si application during salinity stress on the DNA methylation level in oat plants. The results of this study indicated that the exogenous application of silicon improved the tolerance of the oat plants to salinity. The doses of 0.1% and 0.2% Optysil had the greatest effect on alleviating the impact of salt stress on the oat plants. In this research, the epigenetic as well as the physiological response of plants to the applied experimental factors were analyzed, which is a broad coverage of the research topic on the effects of salinity and silicon on plants.

**Keywords:** *Avena sativa* L.; chlorophyll fluorescence; gas exchange; methylation-sensitive amplified polymorphism (MSAP); plant stress



## 1. Introduction

The oat plant (*Avena sativa* L.) is an annual plant belonging to the *Poaceae* family. In 2020, the global area harvested for oats was over 9.7 million ha, and the production quantity of grain was 25 million tones [1]. The importance of oats in the world is small compared to other cereal species, but the popularity of oats is increasing due to their many possibilities of use. The species is distinct among cereals due to its multifunctional characteristics and nutritional profile [2–4]. It can be used as a health-promoting food for humans, an ingredient of animal feed, and a raw material for the energy industry. Oats are cultivated all over the world and are an important part of the diets of people in many countries because they are a rich source of nutrients [5]. It has been proven that oats and oat byproducts are helpful in preventing and treating lifestyle diseases. The health-promoting effect of oats is primarily due to the high content of soluble fiber in the grain. Nowadays, oats are among the richest and most economical sources of soluble dietary fiber [6]. The soluble fiber from oatmeal and oat bran is very effective at lowering blood cholesterol and normalizing blood sugar levels. Moreover, oat grains are a good source of B complex vitamins, protein, and many compounds that exhibit antioxidant activity [2,5,7–10].

*Avena sativa* L. has phytosanitary properties. Oat roots produce saponins (mainly avenacine) that prevent the development of fungal diseases [11,12]. Furthermore, due to

its developed root system, oat has low nutritional requirements [13]. These properties make it possible to reduce the use of fertilizers and chemical plant protection products in agriculture. New opportunities for the use of oats are currently associated with the use of grain not only for fodder and consumption purposes but also as an industrial plant for energy production [14,15]. Oat grain is characterized by the highest calorific value among cereals. Scientific research reports that the use of oats in the energy sector is economically profitable [16–18]. It also provides environmental benefits, as it reduces the consumption of fossil fuels and the emission of noxious gases and dust into the atmosphere [16,19]. This is a particularly important topic in light of the current energy crisis in the world. Oats as a cultivated species fit perfectly into the idea of sustainable development [20].

Abiotic stresses, such as drought, cold, salinity, and heat, negatively influence the survival, biomass production, and yield of staple food crops [21]. Worldwide, soil salinity is thought to have a negative impact on approximately 800 million hectares of arable land [22]. Soil salinity has a negative impact on plant productivity and is one of the main environmental stressors that inhibit the growth and development of crops due to physiological and biochemical changes in plants. The presence of salt in soil solution reduces the ability of a plant to take up water, and this leads to a reduction in the growth rate [23]. This is referred to as the osmotic or water-deficit effect of salinity Moreover, if excessive amounts of salt enter the plant in the transpiration stream, there will be an injury to cells in the transpiring leaves, and this may cause further reductions in growth [21]. A high concentration of salt in soil causes osmotic stress and disturbances in ion homeostasis [24–26]. As a result of osmotic stress, the level of reactive oxygen species (ROS) increases in plants and, consequently, the occurrence of oxidative stress, too [27,28]. Furthermore, salinity stress increases chromosomal aberrations, causes MDA and proline accumulation, and severely hampers the AsA–GSH cycle function [29]. High salt levels inhibit the activity of enzymes involved in photosynthesis [30,31]. For crop improvement against salinity stress, it is important to understand the physiological aspects of tolerance to salinity in plants [32]. Photosynthesis is one of the most important biochemical pathways by which plants convert solar energy into chemical energy and grow. It is the most fundamental and intricate physiological process in all green plants and is also severely affected by environmental stresses. Since the mechanism of photosynthesis involves various components (photosynthetic pigments and photosystems, the electron transport system, and $CO_2$ reduction pathways), any damage at any level caused by stress may reduce the overall photosynthetic capacity of a green plant [21,33,34]. Analysis of chlorophyll fluorescence parameters is a useful technique to mirror the condition of plants [35–37].

Numerous studies conducted on different species of crops prove the efficiency of micronutrients in creating plant resistance to environmental stresses [38–40]. One of the possibilities for limiting the negative influence of environmental factors on plants is foliar silicon supplementation. Si is not considered essential for plant growth and development; however, increasing evidence in the literature shows that this metalloid is beneficial to plants, especially under stress conditions [41,42]. Si fertilization could provide farmers with a quick and cheap method for improving crop yield [43]. The alleviation role of silicon in salt stress tolerance has been described in various crops, such as rice [44], sorghum [45], wheat [46], and soybean [47]. Spraying plants with silicon compounds enhances root and plant growth, yield, and quality in monocots as well as dicots in different soil types both under normal and stress conditions [48]. Sprays with silica nanoparticles have some positive effects on growth and yield and are capable of decreasing the infection rate. Foliar application of silicon nanoparticles in *Cymbopogon flexuosus* caused of increase in plant height, dry weight, and leaf area under salinity [23]. Silicon supplementation was beneficial in alleviating the adverse effects of salinity on the physiological and biochemical characteristics of sweet pepper, the growth of plants, and fruit yield [49].

The versatility of the possibilities of using oats, the global problem of saline soils, and the positive role of silicon in mitigating the negative impact of the environment on plants make it necessary to conduct research on these issues.

DNA methylation is one of the key epigenetic mechanisms among eukaryotes that can modulate gene expression without changes to the DNA sequence [50,51]. There are more and more reports indicating that DNA methylation plays a vital role in turning gene expression in response to environmental stimuli [52]. Many convenient methods to detection epigenetic mechanisms and control gene expression in relation to stress responses are being developed. One of the most popular techniques used in the detection of DNA methylation levels is methylation-sensitive amplification polymorphisms (MSAP).

MSAP techniques are based on methylo-sensitive primers. In this technique, isoschizomers *Msp*I and *Hpa*II are used as 'frequent-cutter'. Both *Msp*I and *Hpa*II recognize the same restriction site (5′CCGG 3′) but show differential sensitivity to DNA methylation [53]. MSAP is very useful for providing cost-effective quantitative estimates of DNA methylation density. It was used in reactions to salinity stress conditions of cereal plants such as wheat [54–56]; barley [57]; maize [58]; and rice [59].

The aim of this study was to evaluate the effect of silicon foliar application on the physiological and epigenetic reaction of oat (*Avena sativa* L.) under salt stress. A scientific hypothesis was adopted that silicon has a positive effect on the response of oat plants grown in conditions of increased soil salinity.

## 2. Materials and Methods

### 2.1. Material of Plant and Conditions of Growth

The pot experiment was carried out at the University of Rzeszow (Poland). In 10 dm diameter pots in which 1.5 kg of soil with a grain size of clay sand that were placed with a slightly acidic pH, seeds of oat of the cv. Bingo (from Hodowla Roślin Strzelce sp. z o.o., Poland) were sown. The content of the compounds in the soil was at an average level. The experiments were carried out in a growth chamber (model GC-300/1000, JEIO Tech Co., Ltd., Seoul, Republic of Korea) at a temperature of $22 \pm 2$ °C, $60 \pm 3\%$ RH, and a photoperiod of 16:8 h light:darkness.

An aqueous solution of NaCl with a concentration of 200 mM in a volume of 50 cm$^3$ was applied to the soil in each pot in the stage of the first pair of leaves. After 7 and 14 days from the application of the NaCl solution to the soil, foliar application of the foliar fertilizer Optysil was applied (contents 200 g·L$^{-1}$ SiO$_2$). Preparation included three concentrations of 0.05, 0.1, and 0.2% Optysil. Plants in pots without the addition of NaCl and Si were used as the control. Spraying was performed with a laboratory hand sprayer. This was applied via a uniform spraying procedure, and plants were sprayed until they were dripping. At the same time, deionized water to the control pots was applied. Physiological measurements were taken two and seven days after each application of Optysil. The scheme of the experiment is shown in Figure 1.

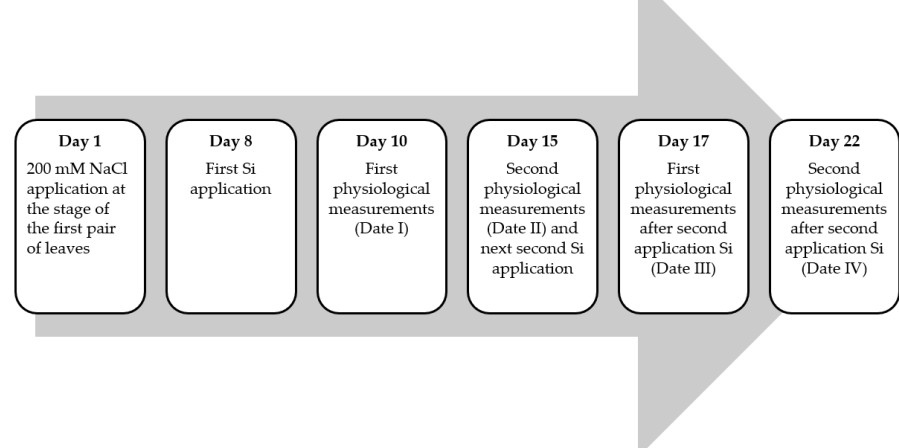

**Figure 1.** The scheme of the time of NaCl application, Si spraying, and physiological measurements in pot experiment.

### 2.2. Physiological Measurements

Measurement of the relative content of chlorophyll (CCI) and chlorophyll fluorescence (maximum quantum yield of primary photochemistry—$F_v/F_0$; maximal photochemical efficiency of PSII—$F_v/F_m$; the performance index—PI; and the total number of active reaction centers for absorption—RC/ABS) and gas exchange (stomatal conductance- $g_s$; net photosynthetic rate—$P_N$; transpiration rate E; and intercellular $CO_2$ concentration—$C_i$) were conducted according to the methods described in publication [60].

### 2.3. Assessment of Fresh Mass and Plant Condition

The visual assessment of the plants' condition and fresh mass content at the final stage of the experiment was assessed. The assessment was carried out on a 9° scale determining the degree of leaf damage. The fresh weight (FM) was determined by weighing the aboveground parts of the plants. The methods are described in detail in the publication [60].

### 2.4. Methylation-Sensitive Amplification Polymorphism (MSAP) Assay and Methylation Data Analysis

MSAP was performed according to Xiong et al. [61] and Xu et al. [53] with some modifications. The genomic DNA (500 ng) of each sample was digested with *Eco*RI/*Msp*I (Thermo Scientific) and *Eco*RI/*Hpa*II (Thermo Scientific) restriction enzymes. The enzymes *Msp*I and *Hpa*II are isoschizomers that recognize the same DNA sequence 5′CCGG 3′. The capacity to cleave at the recognized sequence is dependent on the methylation state of the external or internal cytosine residues. *Hpa*II is inactive if one or both cytosines are fully methylated (both strands methylated; symmetric methylation) but cleaves the hemimethylated sequence (only one DNA strand methylated) whereas *Msp*I cleaves 5′C$^m$CGG 3′ but not 5′$^m$CCGG 3′ [61]. The products of restriction reaction were subjected to ligation with *Eco*RI and *Msp*I or *HpaII* specific adapters (Genomed) to avoid reconstruction of restriction sites at 20 °C for 90 min.

Subsequently, the ligated DNA was diluted to 1:10 and preamplified using *Eco*RI (Genomed) and *Msp*I or *Hpa*II (Genomed) primers with one selective nucleotide at the 3′ ends. The PCR conditions were as follows: 94 °C–30 s, 46 °C–1 min, and 72 °C–1 min for 30 cycles and final extension at 72 °C for 5 min. The preamplified product was diluted 10× with TE buffer and subsequently selectively amplified with different combinations of *Eco*RI and *Msp*I or *Hpa*II (Genomed) MSAP primers (Table 1). Each of the selective primers was equipped with 2–3 selective nucleotides at the 3′ end.

For selective amplification the PCR, conditions were as follows: 94 °C–1 min, 65 °C–1 min (in each successive cycle, the temperature was reduced by 0.7 °C), and 72 °C–1 min for 11 cycles and 94 °C–1 min, 56 °C–1 min and 72 °C–1 min for 23 cycles and final extension at 72 °C–7 min for 1 cycle. To the PCR products, equal volume of formamide dye was added and subjected to electrophoretic separation on 6.0% denaturing polyacrylamide gel. The gels were stained with silver nitrate [62] and further scanned for data recording.

MSAP techniques were analyzed according to Xu et al. [53]. The MSAP bands gel template was transformed into a binary character matrix using '0' and '1' to indicate the absence and presence, respectively, of particular loci. DNA methylation event was observed when bands present in the gel from the reaction *Eco*RI + *Msp*I (M) were absent from the reaction *Eco*R I + *Hpa*II (H). In this case, internal cytosine of 5′ CCGG 3′ sequence was methylated (5′CmCGG 3′) ('symmetric or fully methylation' defined). Simultaneously, presence of band in H and absence in M simultaneously, indicated that the external cytosine of one DNA 5′ CCGG 3′ sequence strand was methylated (5′mCCGG 3′). This is determined as the 'hemimethylated state'.

Percentage methylation was calculated according to Xiangqiana et al. [63] as below:

Methylation (%) = (number of methylated bands/total number of bands) × 100.

**Table 1.** Sequences of adapters and primers used for MSAP analysis.

| MSAP Stage | Primer/Adapter | Sequence |
|---|---|---|
| Ligation | *Eco*RI-Adapter | 5′ CTCGTAGACTGCGTACC 3′<br>3′ CATCTGACGCATGGTTAA 5′ |
| | *Msp*I-*Hpa*II-Adapter | 5′ CGACTCAGGACTCAT 3′<br>3′ TGAGTCCTGAGTAGCAG 5′ |
| Preamplification | Pre-*Eco*RI | 5′ GACTGCGTACCAATTC 3′ |
| | Pre-*Msp*I-*Hpa*II | 5′ GATGAGTCCTGAGTCGG 3′ |
| Selective amplification | *Eco*RI-ACT | 5′ GACTGCGTACCAATTCACT 3′ |
| | *Eco*RI-AG | 5′ GACTGCGTACCAATTCAG 3′ |
| | *Eco*RI-AC | 5′ GACTGCGTACCAATTCAC 3′ |
| | *Eco*RI-AT | 5′ GACTGCGTACCAATTCAT 3′ |
| | *Msp*I/*Hpa*II-ATG | 5′ GATGAGTCCTGAGTCGGATG 3′ |
| | *Msp*I/*Hpa*II-CTC | 5′ GATGAGTCCTGAGTCGGCTC 3′ |
| | *Msp*I/*Hpa*II-CAT | 5′ GATGAGTCCTGAGTCGGCAT 3′ |
| | *Msp*I/*Hpa*II-CT | 5′ GATGAGTCCTGAGTCGGCT 3′ |
| | *Msp*I/*Hpa*II-GT | 5′ GATGAGTCCTGAGTCGGGT 3′ |
| | *Msp*I/*Hpa*II-CA | 5′ GATGAGTCCTGAGTCGGCA 3′ |

*2.5. Statistical Analysis*

Statistical analysis was carried out using TIBCO Statistica 13.3.0 (TIBCO Software Inc., Palo Alto, CA, USA). In order to detect departures from a normal distribution at $p = 0.05$, the Shapiro–Wilk test was performed. The homogeneity of variance was checked. Two-way repeated measures ANOVA were then performed (with the assessment of time as a factor). In order to verify the relationship, Tukey's post hoc test was performed with a significance level of $p \leq 0.05$.

**3. Results and Discussion**

*3.1. Relative Content of Chlorophyll*

The used experimental factors influenced the content of chlorophyll in the oat leaves. Chlorophyll is one of the main characteristics of plant health [64]. Of the different physiological attributes, the chlorophyll content index is one of prime importance in screening crop plants for salt tolerance [65]. Numerous scientific studies report on the negative effect of salinity on the content of chlorophyll in leaves. Salinity caused a decrease in the content of chlorophyll, e.g., in basil [66], cotton [67], and rice [68]. In our studies, the addition of NaCl decreased the CCI in the oat plants (Figure 2). Similarly, in other studies conducted on oats [69–72], a decrease in chlorophyll content under salt stress was noted. Reducing the content of chlorophyll under salinity is a commonly reported phenomenon, and in various studies, chlorophyll concentration has been used as a sensitive indicator of the cellular metabolic state [73]. The decrease in chlorophyll may be due to different reasons, one of them being related to membrane deterioration or inhibition of the ribulose-1,5-bisphosphate enzyme and the structural destruction of the chloroplast and photosynthetic apparatus [74,75]. The addition of silicon fertilizer improved the CCI value. The highest values were at doses of Optysil of 0.1% and 0.2%. Foliar spraying at the lowest concentration (0.05% Optysil) was more effective on the second day after the run compared to the seventh day (Figure 2). The improvement in the content of chlorophyll due to the use of foliar silicon fertilizer under stress conditions is also noted in research by other researchers [76,77]. Si application increases the content of pigments by suppressing the reducing sodium ion toxicity and maintaining chloroplast structure and function. Si in plants provides rigidity and erectness to the leaves, allowing them to receive more light for photosynthetic activities,

further increasing the formation of chlorophyll pigments [25,75,78,79]. In plants treated with salt only and plants from a NaCl plus Optysil 0.05% variant, along with the duration of the experiment, significant decreases in the measured parameter were noted. The lowest values were recorded on the last day of measurements (Date IV). They were lower by 25 % compared to the results obtained at the beginning of the experiment (Date I) (Figure 2).

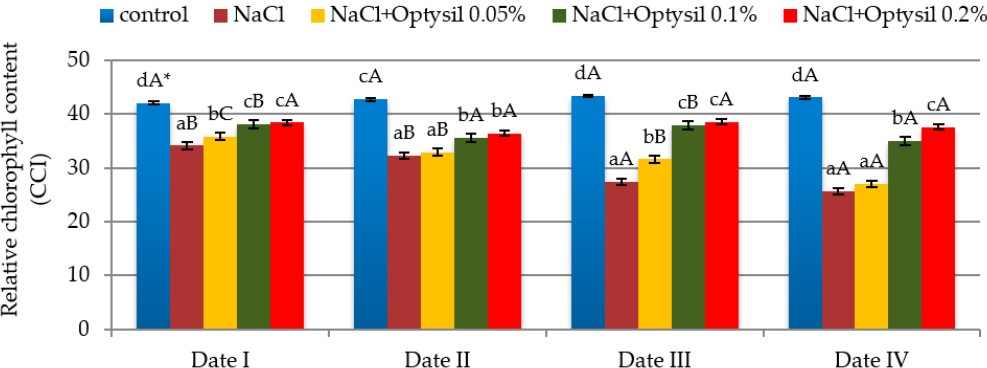

**Figure 2.** The effect of factors of experiment and measurement date on the chlorophyll content in the oat leaves (CCI); (Date I and Date II, 2 and 7 days after first Si application; Date III and Date IV, 2 and 7 days after second Si application); statistical data are expressed as mean ± SD values. * Capital letters show significant differences between the means in the measurement dates, and lowercase letters show significant differences between the means at next measurement dates ($p$ = 0.05).

### 3.2. Chlorophyll Fluorescence

The chlorophyll content and chlorophyll fluorescence are important markers for the photosynthetic health of a plant [23]. Chlorophyll fluorescence analysis has become one of the most powerful and widely used techniques available to plant physiologists and ecophysiologists [80]. Chlorophyll fluorescence measurements have a wide range of applications from a basic understanding of photosynthesis functioning to plant environmental stress responses and direct assessments of plant health [81]. It is an effective method to analyze the health and photosynthetic capacity of plants under normal or stressful conditions (such as salinity) [82]. Photosystem II (PS II) is a relatively sensitive component of the photosynthetic system with respect to salt stress [83]. The treatment of plants with NaCl at the level of 200 mM reduced the parameters of chlorophyll fluorescence (Figure 3). The obtained results confirm the research of other researchers on the negative effect of high salt concentration on the fluorescence of oat plants [84,85]. Salt-affected oat plants might have higher chlorophyllase activity inhibiting chlorophyll biosynthesis and altering chloroplast ultrastructure through oxidative peroxidation [23,86]. Salinity stress hampers general plant growth and affects gas exchange parameters, chlorophyll fluorescence, and nitrogen metabolism [87]. The lowest values of $F_v/F_0$, $F_v/F_m$, PI, and RC/ABS were recorded in plants with the NaCl variant. The Optysil fertilizer in each of the applied concentrations had a positive effect on the analyzed parameters. In the research by Maghsoudi et al. [88] and Ghassemi-Golezani and Lotfi [89], foliar application of silicon also had a beneficial effect on the parameters of chlorophyll fluorescence under salt stress. The highest results in comparison to the control were obtained in plants with the 0.1% Optysil and 0.2% Optysil variants. The positive effect of 0.05% concentration was observed primarily two days after use (Date I and Date III). This may indicate a short-term effect of a low dose of Si. In plants with only NaCl, a statistically significant decrease in PI was noted along with the exposure time of plant salinity (Figure 3C).

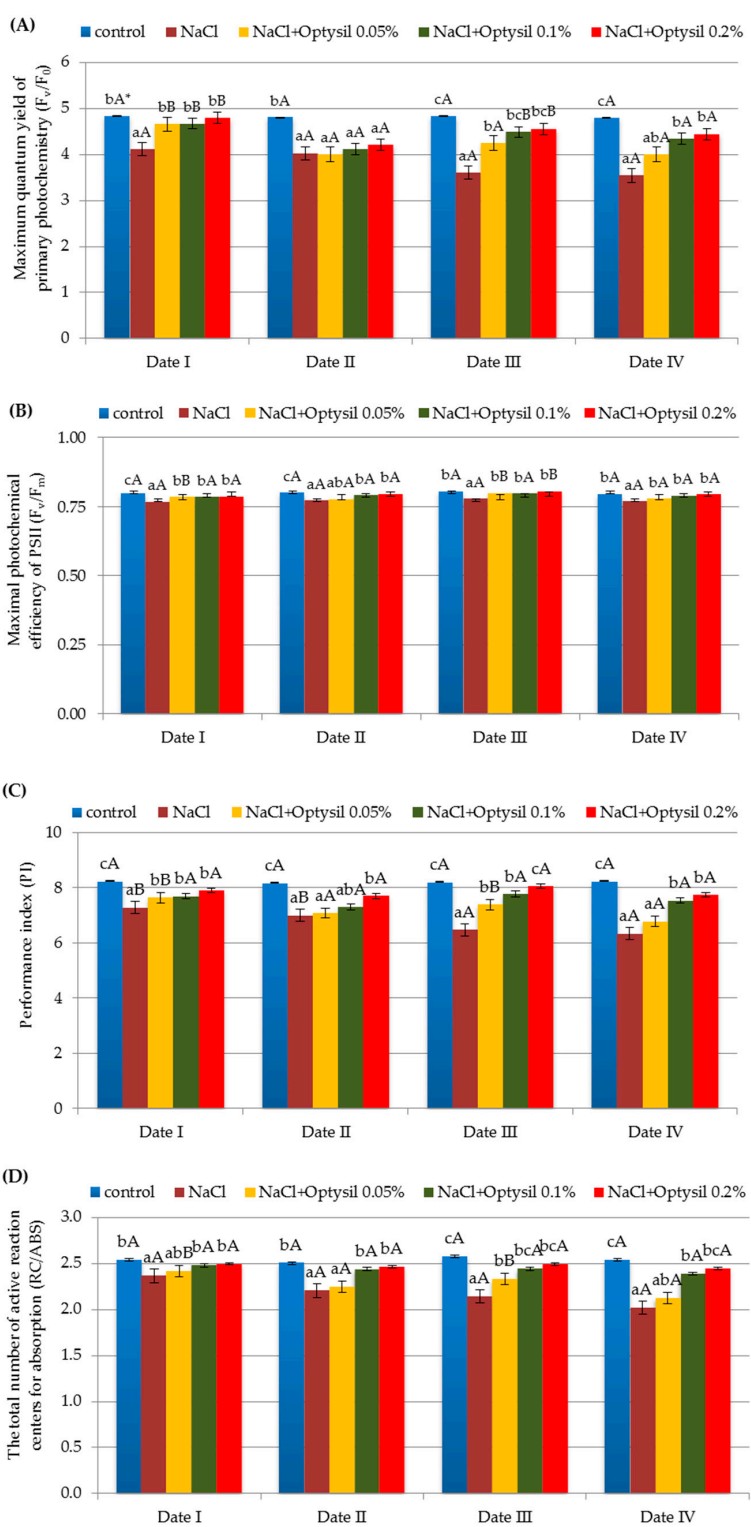

**Figure 3.** The effect of experiment and measurement date on chlorophyll fluorescence parameters: maximum quantum yield of primary photochemistry ($F_v/F_0$) (**A**), maximal photochemical efficiency of PSII ($F_v/F_m$) (**B**), the performance index (PI) (**C**) and the total number of active reaction centers for absorption (RC/ABS) (**D**) in oat plants (Date I and Date II, 2 and 7 days after first Si application; Date III and Date IV, 2 and 7 days after second Si application); statistical data are expressed as mean $\pm$ SD values. * Capital letters show significant differences between the means in the measurement dates, and lowercase letters show significant differences between the means at next measurement dates ($p = 0.05$).

### 3.3. Gas Exchange

In the experiment, the values of the gas exchange parameters depended on the NaCl and the concentration of Optysil. Treatment of the plants with NaCl significantly reduced the $g_s$, $P_N$, E, and $C_i$ values (Figure 4). In the studies conducted on oats by Qin et al. [70] and Shah et al. [90], also as a result of the action of salt, a decrease in the gas exchange parameters were noted. Salt stress can lead to stomatal closure, which reduces carbon dioxide availability in the leaves and inhibits carbon fixation, exposing chloroplasts to excessive excitation energy, which in turn increases the generation of ROS and causes oxidative stress [21]. Salinity limits transpiration and gaseous exchange by distorting chloroplast ultrastructure and the PSII system, therefore reducing stomatal conductance [91]. Inhibition of plant assimilation under the influence of salt stress may be caused by a limited supply of $CO_2$ due to the partial closure of the stomata and an impaired biochemical binding process for $CO_2$ [92]. This situation causes the reduction of the photosynthetic binding of $CO_2$, a change in the cellular metabolism, and an increase in the production of ROS in chloroplasts. ROS can damage the photosynthetic apparatus, especially PSII, causing photoinhibition due to an imbalance in the photosynthetic redox signaling pathways and inhibition of PSII repair [93,94]. Generally, plants close the stomata when stressful conditions arise in order to conserve water and in sequel, to reduce stomatal conductance and photosynthesis [95]. In our study, the value of $g_s$ decreased with the duration of the experiment. The addition of silicon increased the $g_s$ value compared to plants with only NaCl. However, in a study by Qin et al. [70], fertilization with silicon caused a decrease in $g_s$ with an increase in the dose of Si. The lowest values in plants subjected to salinity stress were recorded during the third and fourth measurements (Date III and Date IV). The use of Optysil improved the value of the measured parameters (Figure 4). The positive effect of the action was demonstrated at the highest applied foliar fertilizer concentrations of 0.1% and 0.2%. The positive role of the concentration of 0.5% was primarily two days after the application of the spray (Date I and Date III). The application of silicon fertilizer in salinity conditions also caused a significant increase in photosynthesis and the transpiration index in oat plants in the study by Kutasy et al. [77]. The protective role of Si in the photosynthetic apparatus and increased photosynthetic activity can partially be attributed to the greater ability of plants to take up K+ and the enhanced antioxidant defense [96].

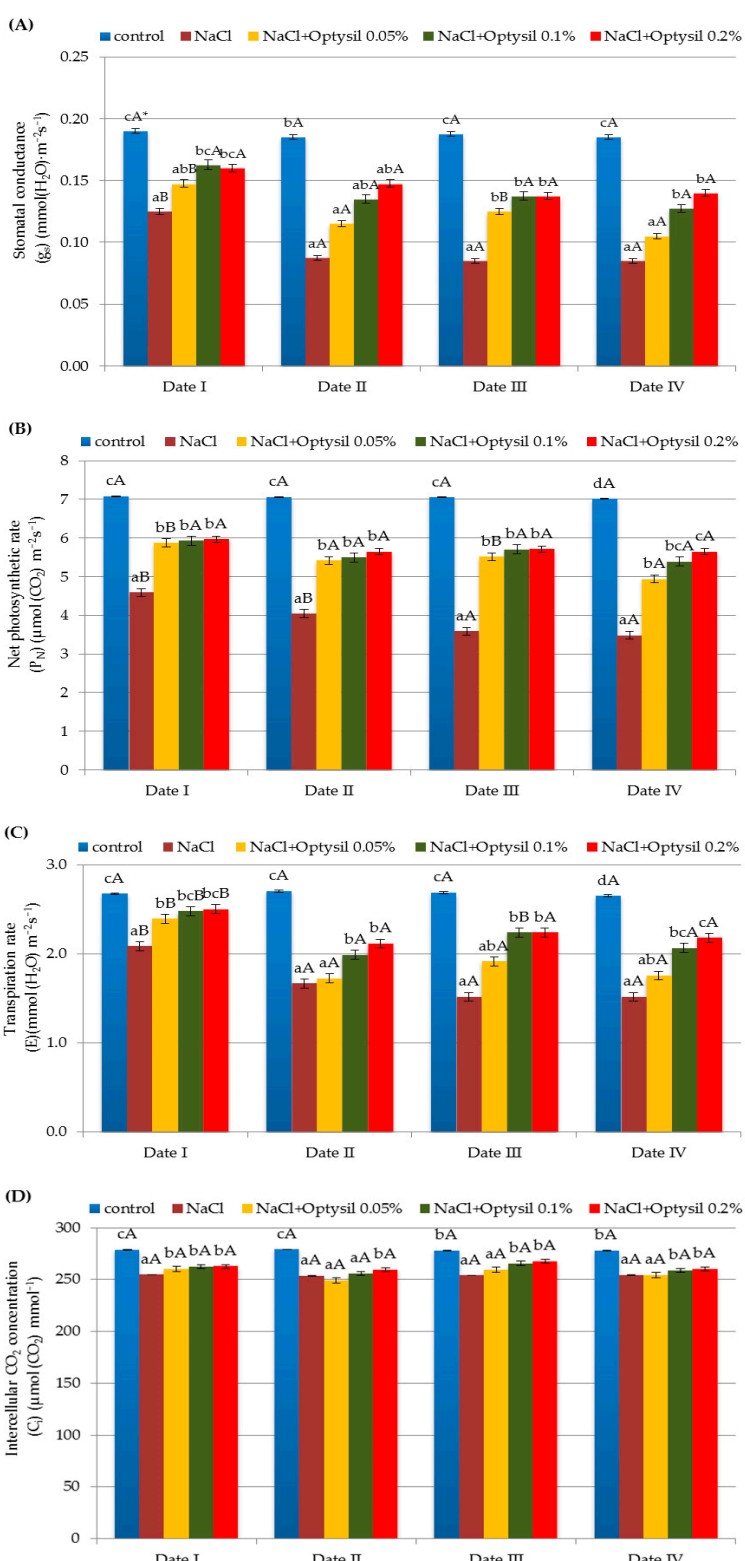

**Figure 4.** The effect of factors of experiment and measurement date on gas exchange parameters: stomatal conductance ($g_s$) (**A**), net photosynthetic rate ($P_N$) (**B**), transpiration rate (E) (**C**) and intercellular $CO_2$ concentration ($C_i$) (**D**) in oat plants (Date I and Date II, 2 and 7 days after first Si application; Date III and Date IV, 2 and 7 days after second Si application); statistical data are expressed as mean ± SD values. * Capital letters show significant differences between the means in the measurement dates, and lowercase letters show significant differences between the means at next measurement dates (*p* = 0.05).

### 3.4. Fresh Mass and Plant Condition

In an experimental setting, one of the first observable responses of plants after salinity exposure is a reduction in shoot growth [97,98]. This dependence is observed in various species of cultivated plants [99–102]. In the final stage of the pot experiment, the fresh mass of the plants (FM) was estimated, and the general condition of the plants was assessed. It was observed that 200 mM NaCl had a negative effect on plant growth. Plants treated with salt showed a lower content of FM and were characterized by a worse visual condition (Figure 5). In plants treated with NaCl only, FM was lower by 31.7% compared to the control plants. The obtained results are consistent with reports on oat by other researchers [69,71,103]. Salinity negatively influences leaf expansion and water levels. Imbalances in the plant water status, turgor reduction, and stomatal closure cause growth inhibition through the reduction in photosynthesis [104]. In our study, the inhibition of vegetative growth in plants submitted to salinity can be associated with a marked inhibition of photosynthesis. Optysil foliar application improved FM. The best effects were noted at 0.1% and 0.2%. In these variants, the FM content was 18.9% and 22.1% higher, respectively, compared to the plants without the addition of Optysil (Figure 5).

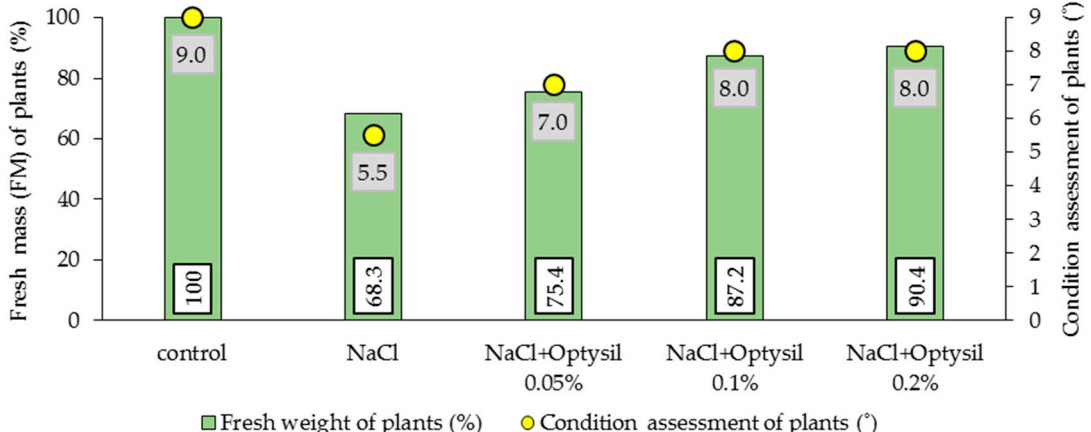

**Figure 5.** The effect of salinity and Si dose on the amount of fresh mass (FM) and condition of plants.

### 3.5. DNA Methylation Level

MSAP techniques based on methylo-sensitive primers were performed in the presented research. This method can detect substantial differences in the proportion of methylated cytosines in 5′CCGG 3′ genomic regions depending on cytosine position.

To detect the frequency of methylation, seven combinations of selective primers were used. As a result, 321–340 total products of selective amplification were obtained (Table 2). An electropherogram photo showing the results of DNA methylation profiles is shown in Figure 6. As can be seen, a highly unlikely banding pattern of *Eco*RI + *Msp*I reaction products compared to *Eco*RI + *Hpa*II can be observed (Figure 6).

**Table 2.** The results of methylation analysis.

| Analyzed Values | Control | NaCl | NaCl + 0.05% Si | NaCl + 0.1% Si | NaCl + 0.2% Si |
|---|---|---|---|---|---|
| Number of symmetric methylation bands | 44 | 45 | 48 | 54 | 57 |
| Symmetric methylation (%) | 13.7 | 13.2 | 14.2 | 16.5 | 17.0 |
| Number of hemimethylation bands | 97 | 99 | 90 | 87 | 74 |
| Hemimethylation (%) | 30.2 | 29.1 | 26.6 | 26,6 | 22.1 |
| Total bands number | 321 | 340 | 338 | 327 | 335 |
| % total methylation | 43.9 | 42.4 | 40.8 | 43.1 | 39.1 |

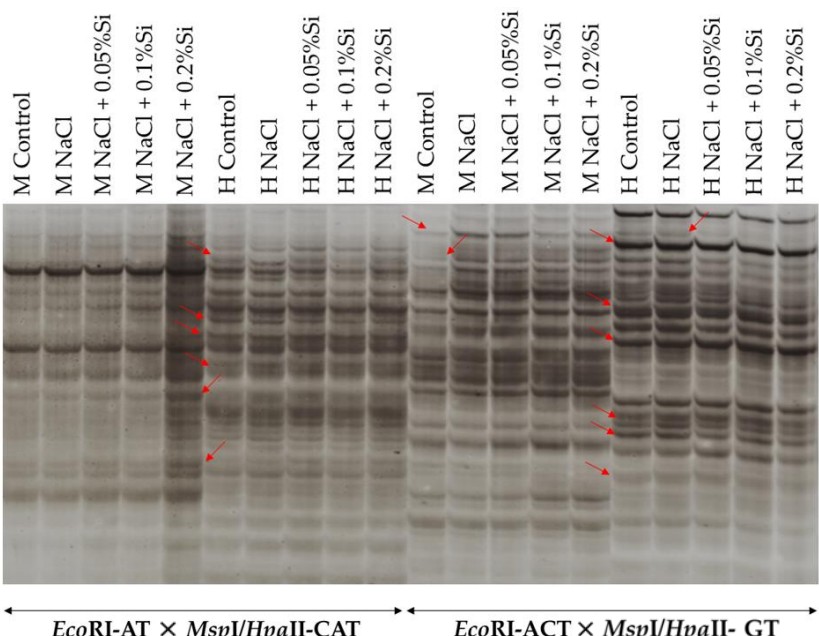

**Figure 6.** Example image of selective amplification products of *Eco*RI-AT × *Msp*I/*Hpa*II-CAT and *Eco*RI-ACT × *Msp*I/*Hpa*II- GT selective starters; Red arrows indicate polymorphic bands.

The methylation frequency (symmetric, hemi-, and total methylation) was calculated. The results are presented in Table 2. Salinity conditions led to a decrease in DNA methylation. This phenomenon was observed in the case of NaCl salinity conditions as well as after the silicon application. Nevertheless, only a slight lowering of methylation level followed by a moderate dose of Si (0.1%) was observed. The highest decrease in total methylation was obtained after 0.2% Si dose application under salinity conditions.

Additionally, in each samples predominance of hemimethylation to symmetric ones was observed. This is especially visible in the case of the control plants. The application of silicon in higher concentrations under salinity conditions led to an increase in symmetric methylation and a decrease in hemimethylation simultaneously. A comprehensive understanding of the mechanism of stress action is important, as it affects the growth of plants not only by altering several crucial plant physiological processes (e.g., photosynthesis, carbon metabolism, mineral nutrition, and oxidative events) but also the plant epigenetic mechanisms [104,105].

DNA methylating/demethylating activities and alteration of the chromatin structure by chromatin remodeling or histone modification interact and influence gene expression. These topics are actively researched and extensively reviewed [106–109]. The latest reports indicated a significant role of the epigenetic mechanism in the reaction of plants in stressed conditions [54,106,110,111]. Scientists prove that epigenetic mechanisms not only allow plants to survive unfavorable environmental conditions but also participate in the formation of 'epigenetic memory'. A pre-exposure (priming) to different types of stress may alter subsequent responses by displaying a faster and/or stronger activation of the various response pathways [105,112–115].

The researchers in [116,117] indicated the relationship between DNA methylation and gene expression. Li et al. [118] indicated that expressed genes during leaf development in Zea may show the lowest levels of methylation. In the present research, oat plants that grew under salinity conditions were characterized by a decrease in methylation level. Similar results were observed in wheat [55], rice [119], and sugar beet [120]. Additionally, Ferreira et al. [119] indicated that salt stress tolerance rice variety characterized a lower level of methylation than the salt stress-sensitive variety. According to results obtained by Skorupa et al. [120], DNA methylation changes may be involved in salt tolerance and transcriptomic response to salinity.

Therefore, the reduced level of methylation in oat plants obtained under salt stress would suggest the activation of genes responsible for coping with stressful conditions. Additionally, silicon application in different doses modulates the reduction of total DNA methylation. Lukens and Zhan [51] implied that a plant genome's response to environmental stress generates novel epigenetic methylation polymorphisms. Novel, stress-induced epigenotypes may contribute to phenotypic diversity and plant improvement.

This may suggest that treating plants under salt stress conditions with preparations containing Si leads to the formation of epigenetic memory and will allow plants to cope better with stress conditions at the same time. Such memory will enable the plants to survive unfavorable environmental conditions caused by salinity in the next round.

## 4. Conclusions

The target of the conducted research was to assess the effect of silicon foliar fertilizer application on oat (*Avena sativa* L.) plants under salt stress. The results confirmed the hypothesis that silicon has a positive effect on the alleviation of salinity stress in oat plants. The outcomes of the conducted research are indicative of the beneficial effect of silicon on the chlorophyll content in leaves and the selected parameters of chlorophyll fluorescence, gas exchange, fresh mass, and the condition of plants. The doses of Optysil 0.1% and 0.2% were the most applicable. The application of Si under salinity conditions influences the decreasing degree of methylation. DNA methylation as an epigenetic mechanism plays a significant role in the reaction of plants to stress conditions. Changes in DNA methylation not only allow plants to survive unfavorable environmental conditions but also participate in the formation of 'epigenetic memory'. The epigenetic memory of oat plants, acquired during growth under salinity stress conditions, and silicon treatment allow plants to cope with stress in a better way for the next round. The obtained results can be used to create a strategy for reducing the negative impact of abiotic stresses on agricultural productivity. The results of research should be verified in the field where various environmental factors, e.g., weather conditions, can modify the reaction of plants to stress conditions and the effect of Si foliar application.

**Author Contributions:** Conceptualization, B.S. and R.T.-S.; methodology, B.S., R.T.-S. and M.M.; formal analysis, B.S.; investigation, B.S., R.T.-S. and M.M.; writing—original draft preparation, B.S. and M.M.; writing—review and editing, B.S. and M.M.; visualization, B.S.; supervision, R.T.-S. All authors have read and agreed to the published version of the manuscript.

**Funding:** This research was funded from the financial resources of the Ministry of Science and Higher Education for scientific activities of the Institute of Agricultural Sciences, Land Management and Environmental Protection, University of Rzeszow.

**Institutional Review Board Statement:** Not applicable.

**Data Availability Statement:** The data presented in this study are available upon request from the corresponding author upon reasonable request.

**Conflicts of Interest:** The authors declare no conflict of interest.

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
