# Peer review of "Effect of Silicon on Oat Salinity Tolerance: Analysis of the Epigenetic and Physiological Response of Plants"

_agriculture, doi:10.3390/agriculture13010081_

Round 1
Reviewer 1 Report
The present manuscript, ‘Effect of silicon on oat salinity tolerance: Analysis of the epigenetic and physiological response of plants’ by Stadnik et al. is a timely study and overall written nicely. I congratulate the authors on taking note of this extreme issue of soil salinity and their painstaking efforts. Nonetheless, I think the article could have been better as the authors do not seem to stretch the full potential of their research. I have a few concerns and suggestions that could assist authors in this.
Major comments:
1. Section 3.3. and 3.5 are represented and discussed nicely. However, I do not see authors discussing other sections. Most of the time, it went, ‘we saw this, and xx et al. also saw this’. It is quite disappointing for me. I would invite authors to take their time and discuss ‘what could have been the reason in their opinion of the observed phenomena and then try to support their understanding with other published studies’.
2. I would include a research perspective, probably at the end of the ‘conclusion’ section.
3. Line 116: My major concern is the osmotic shock here since the authors generated 200 mM salt stress in one go. Ideally, it should have been provided in smaller doses. For example, apply 50 mM every other day until the desired concentration is achieved.
4. Line 134: Assessment of fresh weight might not be the best parameter to look for salinity-implicated restriction as it can fluctuate drastically depending on several other agricultural practices. The authors are strongly suggested to provide dry weights of Oat plants.
Minor suggestions:
1. Line 85: Although both reviews mentioned here were good, especially Epstein et al. 2009, I believe it would be nicer if authors also included recent research on silicon supplementation (better if foliar) both under optimal and stress conditions, e.g. (https://doi.org/10.1016/j.jhazmat.2021.125254, https://doi.org/10.3390/plants9060733) or reviews (https://doi.org/10.1016/j.envpol.2022.119855, https://doi.org/10.1007/s00344-020-10172-7, https://doi.org/10.3389/fpls.2020.01221).
2. Line 119-120: % of what?
3. In continuation to point 4 under the major comments section (see above):
Line 137-139: Again, here, I would advise including root weight and length. It holds special relevance in salinity studies.
4. Figure 1: I’m a bit lost here. So, the second assessment of chlorophyll and fresh weight was done on the same day as the second Si application? For the same reason, I would encourage authors to present maybe an experimental timeline figure. It would help us grasp their study better.
5. In sections 3.2 and 3.3, apart from major concerns (please see the abovementioned suggestions), further salt implications and mechanisms can be discussed a bit more with the support from original research, e.g., https://doi.org/10.1016/j.plantsci.2004.04.020, https://doi.org/10.3389/fpls.2022.903954.
Reviewer 2 Report
Paper Title- Effect of silicon on oat salinity tolerance: Analysis of the epigenetic and physiological response of plants submitted to Agriculture, MDPI.
(Authored by Barbara Stadnik, Renata Tobiasz-Salach and Marzena Mazurek)
Recommendation- Minor Revision
_______________________________________________________________
In this article, the aim of the study was to evaluate the effect of silicon foliar application on the physiological and epigenetic reaction of oat (Avena sativa L.) under salt stress. For this, a Pot experiment was carried out in controlled conditions. Oat plants were subject to sodium chloride (NaCl) at a concentration of 200 mM. Three concentrations of Optysil (200 g∙l-1 SiO2) were used as foliar fertilization. Measurements were made of relative chlorophyll content in leaves, selected chlorophyll fluorescence parameters and gas exchange parameters. In study, methylation-sensitive amplification polymorphisms (MSAP) analysis was used to investigate the effect of Si application during salinity stress, on DNA methylation level in oat. The results of the study indicated that the exogenous application of silicon improved the tolerance of oat plants to salinity.
Reviewer’s comments- the manuscript requires the incorporation of the following points. It includes clearly innovative ideas for further research in the field.
Title- The title is accurate as giving overall idea of the experiment executed.
Abstract-
- Abstract is up to the mark and concisely written with a complete short introduction of the manuscript.
- May be the addition of a line about how your work is different from others in the field would be helpful to the readers.
- Abstract is more about experimental strategy; equal result orientation will balance it up.
Introduction-
· The addition of some new papers from 2021-2022 can enhance the current information in the field. Some of the relevant papers are recommended to add within manuscript.
· Apart from this, the rest introduction is well written.
Materials and Method
· Adding the name of the place where you got the variety would be helpful to the reader.
· In line no. 143 use of by should be avoided for more clarity.
· Following references should be incorporated to strengthen and validate the results of the present experiment.
· Gupta, P., Srivastava, S., and Seth, C. S. (2017). 24-Epibrassinolide and sodium nitroprusside alleviate the salinity stress in Brassica juncea L. cv. Varuna through cross-talk among proline, nitrogen metabolism and abscisic acid. Plant and Soil, 411(1), 483-498.
· Prajapati, P., Gupta, P., Kharwar, R. N., and Seth, C. S. (2022). Nitric oxide mediated regulation of ascorbate-glutathione pathway alleviates mitotic aberrations and DNA damage in Allium cepa L. under salinity stress. International Journal of Phytoremediation, 1-12.
Results and discussion-
- In line no. 195 NaCL, L should be in lowercase.
Conclusion-
· Conclusion is concise, up to the mark and professionally written.
References
· Please check the formatting according to the journal’s requirements. As far as checked following a constant format.
Tables and figures
· The legend of figure 4 should be more elaborative to enhance the reader's understanding.
Round 2
Reviewer 1 Report
The present manuscript, ‘Effect of silicon on oat salinity tolerance: Analysis of the epigenetic and physiological response of plants’ by Stadnik et al., has been revised appropriately. Most of my concerns have been addressed by the authors. The improved discussion is more balanced now. Further, the experimental timeline and research perspectives are added nicely. I have only a few minor corrections at this point:
Line 91: A recent review https://doi.org/10.1016/j.envpol.2022.119855 could be cited here.
Line 96: The citation [48] does not seem befitting here rather, it could be more appropriate in line 66.
Lines 99: Please correct the citation to https://doi.org/10.1016/j.jhazmat.2021.125254